# Breast Cancer Stem Cells as Drivers of Tumor Chemoresistance, Dormancy and Relapse: New Challenges and Therapeutic Opportunities

**DOI:** 10.3390/cancers11101569

**Published:** 2019-10-15

**Authors:** Maria Laura De Angelis, Federica Francescangeli, Ann Zeuner

**Affiliations:** Department of Oncology and Molecular Medicine, Istituto Superiore di Sanità, 00161 Rome, Italy; marialaura.deangelis@iss.it (M.L.D.A.); federica.francescangeli@iss.it (F.F.)

**Keywords:** breast cancer, breast cancer stem cells, tumor dormancy, quiescence, drug resistance, plasticity, tumor heterogeneity, metastasis, targeted therapies

## Abstract

Breast cancer is the most frequent cancer among women worldwide. Therapeutic strategies to prevent or treat metastatic disease are still inadequate although great progress has been made in treating early-stage breast cancer. Cancer stem-like cells (CSCs) that are endowed with high plasticity and self-renewal properties have been shown to play a key role in breast cancer development, progression, and metastasis. A subpopulation of CSCs that combines tumor-initiating capacity and a dormant/quiescent/slow cycling status is present throughout the clinical history of breast cancer patients. Dormant/quiescent/slow cycling CSCs are a key component of tumor heterogeneity and they are responsible for chemoresistance, tumor migration, and metastatic dormancy, defined as the ability of CSCs to survive in target organs and generate metastasis up to two decades after diagnosis. Understanding the strategies that are used by CSCs to resist conventional and targeted therapies, to interact with their niche, to escape immune surveillance, and finally to awaken from dormancy is of key importance to prevent and treat metastatic cancer. This review summarizes the current understanding of mechanisms involved in CSCs chemoresistance, dissemination, and metastasis in breast cancer, with a particular focus on dormant cells. Finally, we discuss how advancements in the detection, molecular understanding, and targeting of dormant CSCs will likely open new therapeutic avenues for breast cancer treatment.

## 1. Introduction

Breast cancer (BC) is the most common cancer in women and the second cause of cancer-related death among women worldwide [1]. Current therapeutic strategies have a limited efficacy on patients who are either metastatic at presentation or experiencing disease recurrence despite significant advancements in BC diagnosis and treatment. Therefore, new knowledge is urgently needed to understand the mechanisms leading to metastatic BC and to devise effective therapeutic strategies. BC has been classified into different subtypes according to distinct gene expression signatures and histological features [2,3] and it is the object of continuous efforts that are dedicated to unravelling the genetic mutations responsible for tumor initiation and metastasis [4,5]. However, BC results from complex interactions between genetic determinants and environmental influences, including lifestyle-related factors. Genetic and environmental factors converge to generate a high degree of heterogeneity that represents an endless source of tumor variability. Heterogeneity manifests between cancers from different patients (inter-tumor heterogeneity) and within a single tumor (intra-tumor heterogeneity) [6]. Latest research using “omics” platforms, such as single cell DNA and RNA sequencing, are opening new scenarios in understanding BC heterogeneity by identifying distinct cell populations that are associated with treatment resistance and metastasis. Outstanding contributions in this field were recently provided by single-cell sequencing studies showing the dynamics of response to neoadjuvant chemotherapy in triple negative BC (TNBC) and the existence of signatures of chemoresistance that are able to predict long-term patient outcomes [7,8]. Cancer stem cells (CSCs) represent, at the same time, a source and a product of tumor heterogeneity. In fact, they contribute to tumor heterogeneity with a high degree of plasticity, resulting in the generation of cells with a variety of phenotypic, functional, and metabolic features. However, simultaneously, they also respond to a plethora of micro- and macro-environmental stimuli, thus reflecting the heterogeneity of the tumor microenvironment [9]. CSCs exploit interactions with the tumor microenvironment to self-renew, resist to radio- and chemotherapy, and generate distant metastases [10,11,12]. In particular, microenvironmental stimuli that are delivered by non-tumoral cells, such as the interaction with niche components and immune system cells, continuously shape and strengthen the CSCs population [13]. CSCs plasticity is particularly evident in the ability of stem cells to oscillate between proliferative and quiescent states to optimize their survival opportunities. Quiescent cells with CSCs features have been demonstrated to resist harsh environmental conditions, escape anticancer treatments, and hide from the immune system [9]. In breast and other tumors, quiescent CSCs are present before therapeutic challenges, accumulate upon radio-chemotherapy, lurk in the bloodstream as circulating tumor cells (CTCs), and persist for up to two decades in premetastatic sites as disseminated tumor cells (DTCs). Thus, quiescence and dormancy represent key properties that characterize the whole lifetime of CSCs, involving molecular mechanisms that have only been partially understood. Understanding the biology of dormancy in BC is instrumental to improve the effectiveness of anticancer treatments and prevent late metastatic relapses that characterize estrogen-receptor (ER)-positive BC. In this review, we summarize the current understanding on quiescent and dormant breast CSCs in tumor chemoresistance, dissemination, and recurrence. Finally, we discuss the clinical relevance of quiescent and dormant CSCs in breast tumors and the potential therapeutic strategies that aimed at improving the metastasis-free survival of BC patients.

## 2. Plasticity of the Breast Cancer Stem Cell Compartment

Breast Cancer Stem Cells (BCSCs) were initially described in 2003 by Al-Hajj and colleagues, who found that the CD44^+^CD24^−/low^Lin^−^ fraction was significantly enriched for cells with tumor forming ability as compared to the CD44^+^CD24^+^Lin^−^ population. Moreover, tumors that formed by CD44^+^CD24^−/low^Lin^−^ cells could be serially passaged and reproduced the cellular heterogeneity observed in the tumor of origin [14]. Subsequently, populations of CD24^+^CD29^+^ cells and CD24^+^CD49f^+^ cells were isolated from BRCA1-mutated mammary tumors and displayed self-renewal and tumor-initiating capacity in vivo [15]. Several other surface proteins have been then indicated as putative markers for BCSCs, including CD133, CD61, CD49f, CXCR4, ANTXR1, integrin-β4 [16,17,18,19,20,21]. Moreover, elevated levels of intracellular proteins linked to stemness, self-renewal, and epithelial-mesenchymal transition (EMT) have been reported to characterize BCSCs, such as sex determining region Y-box2 (SOX2), sex determining region Y-box9 (SOX9), SNAIL, polycomb group RING finger protein 4 (PCGF4/BMI1) and stem-SH2-containing 5′-inositol phosphatase (s-SHIP) [22,23,24,25,26]. Finally, other methods of BCSCs enrichment have been reported, such as aldehyde dehydrogenase (ALDH) activity, side population features, and autofluorescence [27,28,29,30]. Despite the multiplicity of putative BCSCs markers, there is currently no definitive agreement on BCSCs phenotypic characterization or a universal combination of markers that could specifically identify BCSCs in all BC subtypes. Likewise, reports on the clinical significance of BCSCs markers are mainly controversial [31]. Such a situation is not surprising when considered at the light of inter- and intra-tumor heterogeneity, which is likely to translate into BCSCs populations displaying variable phenotypes, molecular profiles, and biological properties. Most importantly, BCSCs phenotype reflects the high degree of plasticity intrinsic to CSCs, which might result in a transient expression of surface markers used for BCSCs isolation. The increasing evidences of BCSCs plasticity point to this population as a dynamic entity being continuously shaped by microenvironmental features, such as interactions with niche elements, tumoral and non-tumoral cells, soluble factors, and anticancer therapies. BCSCs plasticity has several crucial implications for both experimental and clinical oncology. First, the differentiation process that leads to BCSCs loss of self-renewal can happen in the opposite way, which means that non-stem breast cancer cells can become BCSCs. The process of dedifferentiation has been analyzed in BC cell lines [32] and observed to occur both in normal and transformed mammary epithelial cells, which can acquire a stem-like state and enhanced tumorigenicity in vivo [33]. From a clinical point of view, the ability of non-stem cancer cells to acquire BCSCs properties and, in general, the plasticity of the BCSC compartment pose a serious challenge to targeted therapeutic strategies. In fact, targeting a specific population of BCSCs (such as ER-positive BCSCs in ER^+^ BC) often results in disappearance of the therapeutic target or acquisition of resistance through alternative signalling pathways [34]. The SWH/HIPPO (Salvador-Warts-Hippo) pathway has been reported to play a central role in the transition towards a stem cell state in BC, as a gain of the PDZ-binding motif (TAZ) confers self-renewal capacity and BCSCs features to non-stem BC cells [35]. In line with these studies, it was recently shown that Yes-associated protein (YAP)/TAZ-mediated dedifferentiation is mediated by mechanotransduction signals, which results in a modulation of the autophagic pathway. Specifically, the physical properties of the extracellular matrix were shown to translate YAP/TAZ signaling into the transcriptional control of the autophagic flux, resulting in dedifferentiation and acquisition of a BCSCs state [36]. A key property that is related to BCSCs plasticity is their ability to acquire epithelial or mesenchymal features. An early study by Mani et al. reported that the transition of BCSCs from the epithelial to the mesenchymal state was associated with the gain of stem cell properties [37]. These observations were confirmed by pioneer studies by Chaffer et al., showing that BCSCs plasticity is controlled by zinc finger e-box binding homeobox 1 (ZEB1), a key regulator of stem cell plasticity and EMT [38]. Interestingly, ZEB1 has also been recently shown to regulate a population of stem cells with mesenchymal features in TNBC [21], which further supports the importance of plasticity in BCSCs regulation. Although BCSCs acquire enhanced migratory and metastatic features when they enter a mesenchymal state, their plasticity allows for them also to acquire a highly proliferative and chemoresistant epithelial form, as well as a range of intermediate states between the epithelial and the mesenchymal state. In fact, it was observed that, during the epithelial-like state, BCSCs are proliferative, located centrally within tumors, and characterized by high expression of ALDH, while during the mesenchymal-like state BCSCs are quiescent, located at the tumor-invasive front and characterized by a CD24^−^CD44^+^ phenotype [31]. More recently, BCSCs expressing both epithelial and mesenchymal markers were reported to be more tumorigenic than cells that reside in a pure epithelial or mesenchymal state, which suggests that BCSCs residing in a hybrid state are characterized by an enhanced malignant phenotype [23]. Translating these observations into therapeutic directions implies that targeting the molecular factors that are involved in a specific BCSCs state (proliferative/resident/epithelial or slow-growing/migratory/mesenchymal) would probably result in BCSCs reversion to the alternative state and consequent therapy failure. By contrast, targeting factors that are involved in BCSCs plasticity may effectively eradicate this population by preventing its adaptation to changing microenvironments. In line with this hypothesis, interfering with BCSCs plasticity and EMT has been demonstrated as a way to impede the engraftment of metastasis-initiating cells to premetastatic organs. Specifically, interleukin-1β (IL1β)-mediated inflammatory responses at premetastatic sites maintain metastasis-initiating BCSCs in a ZEB1-positive mesenchymal-like state, which prevents them from generating a highly proliferative epithelial progeny and blocking metastatic colonization [39]. The ability of BCSCs to acquire epithelial or mesenchymal features is also associated to states of increased or decreased proliferation that allow for BCSCs to optimally adapt to microenvironmental conditions. Interestingly, the transition of BCSCs from a highly proliferative state to a more quiescent self-renewing state was reported to be regulated by the balance between MYC and HIPPO signaling pathways. In fact, MYC was reported to stimulate growth through the inhibition of YAP/TAZ-driven clonogenic growth and both effects are kept in balance by 5’ adenosine monophosphate-activated protein kinase (AMPK) in response to changes in mitochondrial dynamics [40]. In summary, BCSCs are increasingly recognized as a dynamic cell population that is characterized by the ability to acquire an array of phenotypic, functional, and metabolic states in response to changing microenvironmental cues. Therapeutic strategies that interfere with BCSCs plasticity and lock BCSCs in a single state are beginning to be explored in preclinical settings and they may potentially revolutionize targeted BC treatments.

## 3. Key Pathways in BCSCs: Embryonic Signals and Cell State Transitions

Analogously to what happens in normal stem cells, the self-renewal of BCSCs is regulated by the activation and reciprocal interaction of pathways that are important for embryonic development, such as WNT/β-catenin, Hedgehog, NOTCH, and HIPPO pathways. As a number of reviews examine the role of each pathway in BC in detail, we will limit our discussion to some aspects particularly relevant for BCSCs biology. The role of WNT signalling in the generation of BCSCs was recognized as early as in 2003 by observing that exogenous expression of WNT pathway components expands a population of bipotent epithelial progenitors and it generates mixed-lineage mammary tumours that are composed of basal and luminal cell subtypes [41]. Interestingly, when WNT-driven tumors were deprived of WNT, they activated multiple strategies to restore growth and to relapse, which further highlights the plasticity of BCSCs responses [42]. WNT signalling was also shown to be essential for BCSCs metastatization through a mechanism that involves the recruitment of Periostin expressed on stromal cells at the premetastatic site [43]. Accordingly, the suppression of WNT/β-catenin signaling decreased the ALDH1^+^ and CD44^+^/CD24^low^ population inhibiting tumor growth and metastatic ability [44]. More recently, WNT signalling was shown to maintain BCSCs self-renewal and plasticity through proliferating cell nuclear antigen (PCNA)-associated factor (PAF), which is highly expressed in BCSCs but not in their non-transformed counterparts [45]. Two studies reported that WNT pathway inhibition is particularly relevant for driving BCSCs into a pro-metastatic quiescent state. First, Harper et al. showed that in human epidermal growth factor receptor 2 (HER2)-positive BC WNT signalling activates an incomplete EMT-like dissemination program, which generates a population of prevalently dormant BCSCs with a hybrid TWIST1^hi^E-cadherin^lo^ mesenchymal phenotype. These cells are present in the early phases of tumor formation and they possess the metastasis-initiating capacity typical of BCSCs [46]. Secondly, Malladi and coworkers isolated metastasis-competent cells from early BC and showed they expressed stem cell factors SOX2 and SOX9, which are essential for tumor survival and regeneration at metastatic sites. Metastatic BCSCs expressed the WNT inhibitor DICKKOPF-1 (DKK1), which was used to self-impose a slow-cycling state with broad downregulation of UL16-binding proteins (ULBP; also known as retinoic acid early transcript, RAET) ligands for natural killer (NK) cells and the evasion of NK-mediated clearance [47]. Notably, DKK1 was recently shown to play a dual role in the regulation of WNT signaling and BC metastatization by inhibiting lung metastases, but at the same time by promoting bone metastasis [48]. These observations highlight the complexity of WNT signaling in BC and raise concerns regarding the use of WNT inhibitors as anti-metastatic therapies. Hedgehog (Hh) signaling is implicated in regulating the proliferation, fate determination, and maintenance of stem cell populations in both normal and malignant breast tissues. Hh signaling components patched 1 (PTCH1), glioma-associated oncogene homolog (GLI) 1 and 2 are highly expressed in normal and neoplastic breast stem cells and they were shown to cooperate with BMI1 to modulate the numbers of mammosphere-initiating cells. Specifically, the overexpression of GLI2 resulted in the production of ductal hyperplasia, modulation of BMI1 expression in mammosphere-initiating cells, and altered mammary development in NOD/SCID mice [49]. In line with these findings, the transgenic overexpression of GLI1 induced histologically heterogeneous mammary tumors [50]. Hh signalling was shown to be particularly important for the interactions between BCSCs and cancer-associated fibroblasts (CAFs). In fact, BCSCs have been reported to secrete the Hh ligand Sonic Hedgehog (Shh), which regulates CAFs via paracrine activation of Hh signaling. In turn, CAFs secrete factors that promote the expansion and self-renewal of BCSCs. Accordingly, the in vivo administration of a Hh inhibitor resulted in delayed tumor formation, reduction of tumor stroma, and reduced BCSCs content [51]. In TNBC, BCSCs were recently reported to produce the Hh ligand Smoothened (SMO), which instructs CAFs to support the acquisition of a chemoresistant stem cell phenotype via fibroblast growth factor 5 (FGF5) expression and the production of fibrillar collagen. In a phase I clinical trial, a combination of the SMO inhibitor Sonidegib with docetaxel provided clinical benefit in 3/12 patients with metastatic TNBC, which suggests that Hh pathway targeting might increase the efficacy of chemotherapy [52]. Interestingly, recent studies highlighted a link between Hh signalling, EMT, and the formation of primary cilia, which are nonmotile cell-surface structures that serve as cell signalling platforms. Specifically, BCSCs have been shown to activate EMT programs that induce both primary cilia formation and Hh signaling. The ablation of primary cilia was sufficient to repress Hh signalling and the tumor-initiating features of BCSCs, indicating primary ciliogenesis and Hh as key mechanisms by which EMT programs promote stemness in BC [53]. Recent studies also revealed an interesting link between Hh signalling and tetraspanin-8 (TSPAN8), an integrin-binding surface glycoprotein that plays a role in the mesenchymal-epithelial transition (MET) that is associated with metastatic outgrowth [54]. TSPAN8 interacts with PTCH1 and it inhibits the degradation of the Shh/PTCH1 complex, resulting in SMO translocation to cilia, expression of NANOG, OCT4, ALDHA1, and resistance of BCSCs to chemotherapeutic agents [55]. The NOTCH signalling pathway is an evolutionarily conserved cell-to-cell communication system that is composed of four receptors (NOTCH1-4) and five ligands (JAG1-2, DLL1-3-4). NOTCH signalling has been shown to play a central role in BC pathogenesis and tumor progression, with different receptors and ligands contributing to BCSCs maintenance and expansion. The activation of the NOTCH pathway has also been proposed to be a hallmark of TNBC and to determine the highly invasive and chemoresistant phenotype characteristic of this BC subtype [56]. NOTCH4 was shown to regulate BCSCs activity, as a constitutively activated form of this receptor induced poorly differentiated BCSCs-enriched basal-like tumors [57]. The activation of the hypoxic response through hypoxia-inducible factor 1α (HIF1α) acts a key switch of BCSCs behaviour through the regulation of NOTCH signalling, which provides new insights on how BC cells acquire stemness in hypoxic conditions [58]. NOTCH3 and interleukin-6 (IL6) signaling were shown to be involved in the generation of hormone therapy-resistant, self-renewing CD133^hi^/ER^lo^/IL6^hi^ BCSCs [59]. Trastuzumab-resistant BC cells were reported to increase NOTCH1 expression, which represses phosphatase and tensin homolog (PTEN) levels, resulting in the activation of mitogen-activated protein kinase/extracellular signal-regulated kinase 1/2 (MAPK/ERK1/2) and contributing to maintaining BCSCs survival and tumor-initiating potential [60]. Recently, NOTCH2 has been reported as a key determinant of BC dormancy and bone marrow dissemination [61]. DLL1 was found to be significantly up-regulated in ER^+^ luminal BC and its expression has been associated with poor prognosis in this subtype but not in other BC subtypes [62], highlighting the context-specific effect of NOTCH receptors and ligands. Finally, Death-associated factor 6 (DAXX) was recently found to counteract BCSCs enrichment driven by hormone therapy in ER^+^ BC. Ectopic expression of DAXX reduced stemness gene expression, NOTCH signaling, and BCSCs survival upon endocrine therapy, which suggests that a combination of hormone therapy and DAXX-stabilizing agents may inhibit ER^+^ tumor recurrence [63]. The HIPPO pathway regulates an array of cellular processes implicated in tumorigenesis, including stem cell plasticity and interaction with the tumor microenvironment [64,65]. YAP/TAZ were shown to control cell fate in BC through several mechanisms. First, TAZ activity was found to be increased in poorly differentiated BC tumors and to correlate with BCSCs phenotype, poor prognosis, and metastasis. TAZ knockdown strongly reduced the number of tumor-initiating cells and affected the ability of primary BC cells to form distant metastases [35]. Consistent with these findings, TAZ overexpression conferred tumorigenic and metastatic abilities to non-stem primary human BC cells [66]. A strong link between HIPPO and BC derived from the discovery that large tumour suppressor kinases (LATS) 1 and 2, which are part of the HIPPO pathway, regulate ER ubiquitination and proteasomal degradation. The ablation of large tumor suppressor kinase 1 (LATS) stabilized ERα and YAP/TAZ, thus promoting the luminal phenotype and increasing the number of bipotent and luminal progenitors [67]. Finally, the Otubain-2 (OTUB2) deubiquitinating cysteine protease was recently identified as a cancer stemness and metastasis-promoting factor that deubiquitinates and activates YAP/TAZ. Poly-SUMOylated OTUB2, which was induced by Epidermal growth factor (EGF) and oncogenic KRAS, was shown to bind and activate YAP/TAZ promoting tumor metastasis [68]. Interfering with HIPPO pathway components might lead to new therapeutic strategies for BC. It was recently demonstrated that the treatment of mice bearing BC patient-derived xenografts (PDXs) with a humanized anti-receptor tyrosine-kinase-like orphan receptor 1 (ROR1) monoclonal antibody repressed the expression of BCSCs genes, reduced the activation of Rho-GTPases, YAP/TAZ, and BMI1, and impaired the capacity of BC PDXs to metastasize [24]. Altogether, the emerging knowledge of BCSCs signalling pathways highlights the complexity of networks that support stemness, self-renewal, and tumor-initiating capacity, but at the same time indicates new avenues of therapeutic intervention.

## 4. The BCSCs Drug Resistance Toolbox

Increasing evidence suggests that BCSCs execute multiple drug resistance mechanisms, including the overexpression of ATP-binding cassette (ABC) transporters, increased ALDH activity, enhanced DNA repair mechanisms, reinforced reactive oxygen species (ROS) scavenging, cell death escape, induction of dormancy, autophagy, and possibly other resistance mechanisms that are yet to be characterized. ABC transporters utilize the energy of ATP binding and hydrolysis to transport various substrates across cellular membranes. They play a crucial role in multidrug resistance by regulating the efflux of a wide variety of anticancer agents [69], but are also being increasingly recognized to regulate multiple functions supporting malignant metabolism [70]. In BC, Britton et al. detected an increased the expression of ABCG2 (breast cancer resistance protein, BCRP) in stem cells that are resistant to mitoxantrone as compared to non-stem cancer cells [29]. Recently, a SOX2-ABCG2-TWIST1 axis was shown to promote stemness and chemoresistance in TNBC, further indicating ABC proteins as potential targets for BCSCs eradication [71]. ALDHs are a family of enzymes involved in the oxidation of intracellular aldehydes to carboxylic acids, and ALDH1 was one of the first markers utilized for BCSCs isolation [27]. The increased levels of ALDH family members were correlated with chemoresistance in several studies [72,73,74]. Croker et al. showed that ALDH^hi^ CD44^+^ BC cells are responsible for both chemotherapy and radiation resistance and ALDHs inhibition sensitizes BCSCs to chemotherapy [75]. Several studies have investigated the relationship of ALDHs with BC prognosis yielding variable results, which was possibly due to BC heterogeneity and the possibility that ALDH1 might play specific roles in different BC subtypes. According to this hypothesis, ALDH1 expression was reported to significantly affect the prognosis of luminal type BC, but not of TNBC and HER2-enriched subtypes [76]. Enhanced DNA repair is another mechanism of therapy resistance exploited by BCSCs. Pioneering work by Philips et al. showed that BCSCs have an enhanced DNA repair capacity following irradiation and they generate lower levels of ROS as compared to non-stem BC cells [77]. Subsequent studies reported that BCSCs in human and murine tumors contain lower ROS levels than corresponding non-tumorigenic cells, develop less DNA damage, and are preferentially spared after irradiation. Lower ROS levels in BCSCs were associated with the increased expression of free radical scavenging systems, and pharmacologic depletion of ROS scavengers decreased BCSCs clonogenicity, resulting in radiosensitization [78]. The nuclear erythroid-related factor 2 (NRF2), which is a master regulator of cellular responses to oxidative stress, was found to be preactivated in BCSCs, even in the absence of oxidative damage through a non-canonical interaction with PERK (Protein kinase RNA-like endoplasmic reticulum kinase). Constitutive PERK-NRF2 signalling was shown to protect BCSCs from chemotherapy by reducing the ROS levels and increasing drug efflux, thus candidating as a target to chemosensitize drug resistant cells [79]. Besides potentiated ROS scavenging systems, BCSCs appear to also have an upregulation of DNA repair genes, as emerged by the transcriptional profiling of BCSCs that were isolated from the mammary gland of p53-null mice [80]. BCSCs populations were also implicated in resistance to platinum compounds in a BRCA1/p53-mutated mouse mammary tumor model, being responsible for the development of chemoresistance, clonal evolution, and tumor progression upon chemotherapy treatment [81]. In the last decade, autophagy emerged as an important response to chemotherapeutic agents in CSCs, with important consequences for the development of chemoresistance. Autophagy was initially characterized as a catabolic pathway that was implicated in the regulation of cell survival and dormancy responsible for ensuring energy balance upon nutrient deprivation, stress, pathogen infections, or hypoxia [82]. Subsequent studies linked autophagy with therapy resistance in tumor cells and in CSCs [83]. However, autophagy can function as a double-edged sword by suppressing tumorigenesis in some contexts and promoting tumorigenesis in other conditions [84]. In BC, autophagy has been shown to be essential for BCSCs tumorigenicity [85] and to represent a critical component of the pro-survival strategies that are employed by BCSCs, especially in the context of harsh tissue microenvironments, such as those that are associated with nutrient deprivation, cytotoxic therapies, and metastatic dissemination [84]. Autophagy pathways linked to signal transducer and activator of transcription 3 (STAT3) or transforming growth factor β/SMAD (TGFβ/SMAD) may play different roles in BCSCs subsets, as the inhibition of either pathway inhibited the formation of both epithelial and mesenchymal BCSCs colonies and combination treatment limited tumor growth and reduced BCSCs number [86]. The role of autophagy as part of BCSCs survival strategy is particularly relevant in the premetastatic setting, as disseminated BC cells can persist for years to decades before recurring as highly aggressive secondary lesions. DTCs that are endowed with the ability to survive metastatic dormancy and initiate recurrent metastatic lesions have been recently demonstrated to be BCSCs [87]. During metastatic latency, BCSCs adopt dormancy-associated phenotypes through several mechanisms, including the upregulation of autophagic pathways, which have been shown to be essential for BCSCs survival and metastatization [88]. The mechanisms by which autophagy promotes BCSCs survival at metastatic sites include the ability to bypass apoptotic stimuli and to resist chemotherapeutic insults, i.e., by activating SRC-mediated TRAIL resistance in bone metastases [89] or by increasing DNA repair and p53 through autophagy-related 7 (ATG7) [90]. Recently, the autophagy machinery was reported to support the dormancy of metastatic BC cells by keeping a low expression of 6-phosphofructo-2-kinase/fructose-2,6-biphosphatase 3 (PFKFB3). The targeted depletion of autophagic factors ATG3, ATG7, or p62/sequestosome-1 (which interacts physically with PFKFB3) restores aberrant PFKFB3 expression in dormant BCSCs, which leads to the reactivation of proliferative programs and metastatic outgrowth [91]. Further evidence supporting a mechanistic link between autophagy and metastastic dormancy came from studies showing the involvement of Spleen Tyrosine Kinase (SYK) in EMT required for BC metastasis. SYK was found at high levels in cells that underwent EMT inside cytoplasmic RNA processing depots that are known as P-bodies, and SYK activity was required for autophagy-mediated clearance of P-bodies during MET. Genetic knockout of ATG7 or SYK pharmacologic inhibition with fostamatinib prevented P-body clearance and MET, which inhibits metastatic tumor outgrowth [92]. In addition to regulating BCSCs plasticity and dormancy, autophagy influences tumor immunosurveillance programs in several ways. On one side, autophagy can counteract anti-tumor immune responses that are mediated by natural killer and cytotoxic T lymphocytes [93]. On the other side, autophagy has been also described as an important mechanism that is involved in antigen presentation to T cells and possibly in the maturation of some immune cells [93]. This double role of autophagy in tumor immune regulation might at least explain, in part, the difficulties encountered in the clinical application of autophagy inhibitors, together with the low potency and efficacy of many anti-autophagic drugs tested to date. Recently, new autophagy inhibitors IITZ-01 and IITZ-02 have been reported to display potent antitumor action through autophagy inhibition and apoptosis induction in a TNBC xenograft model, awaiting for future clinical validation [94]. Finally, autophagy is tightly linked to hypoxia signalling and to the hypoxia-induced dormancy program in BC. Hypoxic responses can also be induced by chemotherapy, which promotes a signalling cascade that involves calcium release from the endoplasmatic reticulum and the expression of pluripotency genes, which leads to an enrichment of BCSCs [95]. In summary, there is an emerging overlap between pathways that control BCSCs stemness, chemoresistance, metabolism, and premetastatic dormancy, which can be potentially exploited to inhibit BCSCs reactivation and tumor relapse.

## 5. Quiescent/Slow Cycling Stem Cells in Primary Breast Tumors

BCSCs in a slow cycling state or in a non-proliferating state are present throughout the clinical history of BC patients and crucially influence disease outcome (Figure 1). Therefore, a deeper understanding of quiescent and dormant states and of the underlying molecular mechanisms is necessary to improve the effectiveness of BC clinical management. Cell quiescence is defined as a reversible G0/G1 phase that is characterized by the ability of cells to re-enter the cell cycle in response to physiological stimuli. Quiescence is generally considered a transient state, in contrast to the more stable state of dormancy. Recent discoveries suggest that quiescence is not just a passive state, but rather a finely regulated program that can be triggered in response to new microenvironmental signals or to the absence of cues on which the cancer cells previously depend [9]. In several tumors, such as colorectal cancer, pancreatic cancer, melanoma, and glioblastoma, a partial overlap has been identified between CSCs and quiescent/slow-cycling cells. Quiescent/slow cycling CSCs were characterized by an increased tumor-repopulating ability and by the capacity to survive chemotherapy [96,97,98,99,100]. In pioneer studies regarding human and murine mammary tissues, Pece and coworkers isolated from normal mammospheres a quiescent/slow cycling cell population with stem cell features, pointing to quiescence as a key functional trait of breast stem cells. The same authors identified slowly dividing CSCs in breast tumors and demonstrated that such population is particularly abundant in poorly differentiated G3 cancers based on the transcriptional signature of normal quiescent/slow cycling stem cells. These studies revealed the existence of quiescent/slow cycling BCSCs as a central aspect of tumor heterogeneity [101]. Quiescent/slow cycling CSCs in primary BC and other tumors are tightly linked with the presence of hypoxic, acidic, and necrotic regions. In fact, such nutrient-deprived inhospitable regions have been demonstrated to promote stemness programs together with quiescent and migratory phenotypes [13]. In BC, key insights were recently provided by Fluegen et al., who showed that the primary tumor hypoxic microenvironment contains a subpopulation of tumor cells that are characterized by the combined activation of hypoxia (glucose transporter 1/GLUT1, HIF1α) and dormancy (nuclear receptor subfamily 2 group F member 1 NR2F1 and p27) genes. Post-hypoxic cells became chemoresistant DTCs, which, in ER^+^ BC, were dependent on N2RF1-mediated dormancy [102]. Interestingly, BCSCs of ER-positive and negative BC subtypes have been reported to respond differently to hypoxia, as hypoxic exposure induced an increase in BCSCs dependent on estrogen and NOTCH signalling in ER-positive cancers, but a BCSCs decrease in ER-negative cancers [103]. The interactions between hypoxia and quiescence remain mainly unexplored despite these insights, due at least in part to the technical challenges of modelling quiescence in vitro and in vivo. An important contribution to this field recently came from the establishment of an in vitro model of hypoxia/reoxygenation, which allowed for identifying cells surviving hypoxic stress as a dormant BCSCs population with CD24^−^/CD44^+^/epithelial surface antigen (ESA)^+^ expression and spheroid forming capacity [87]. Other investigators established a hypoxia-sensing xenograft model to identify hypoxic tumor cells in vivo with the GFP reporter. This system allowed for isolating ex vivo hypoxic tumor cells, which displayed BCSCs features and enhanced activation of the PI3K pathway [104]. Interestingly, the hypoxic BCSCs phenotype was relatively stable following in vivo re-implantation. This observation is consistent with studies by Fluegen et al. that showed DTCs derive from hypoxic cells in primary tumors, which suggests that BCSCs formed in hypoxic areas may have a particularly stable quiescent phenotype. Finally, an acidic tumor microenvironment has been reported to be associated with poor patient prognosis, resistance to radio- and chemotherapy, and immune suppressive features [105]. Although the relationships between acidic microenvironment and cancer stemness are largely unknown, some evidences indicate that the acidic microenvironment might provide a supportive niche for dormant tumor cells, therefore supporting DTCs survival and metastasis formation [106].

## 6. Disseminated Tumor Cells: Dormant BCSCs in Premetastatic Niches

A population of drug-tolerant cells has been identified in several tumors upon treatment with chemotherapeutic or targeted agents [107,108,109,110,111]. Drug-tolerant cells, also named drug-tolerant persisters (DTPs), are characterized by a transient non-mutational state that allows for them to survive chemotherapeutic insults [112]. In fact, drug tolerance is a temporary condition that can revert when the drug is removed from the culture [108,113]. However, in the presence of continuous drug stimulation, DTPs can evolve into more stable drug resistant populations [107,108]. DTPs represent a variable percentage (0.2–5%) of the parental cancer cell population and they have been identified as largely quiescent cells [108]. In BC, drug-tolerant subpopulations have been shown to possess stem cell features and to depend on ALDH for protection from reactive oxygen species and survival [114]. Recently, Hangauer and coworkers demonstrated that DTPs that arise upon lapatinib treatment of a HER2-amplified BC cell line displayed CD133 and CD44 upregulation and antioxidant-gene downregulation, showing a specific dependency on lipid hydroperoxidase GPX4 (glutathione peroxidase 4) for survival. The loss of GPX4 function resulted in selective DTPs ferroptotic death and prevented tumour relapse in mice, which suggests that targeting of GPX4 may represent a therapeutic strategy to prevent acquired drug resistance [115]. The involvement of epigenetic mechanisms rather than genetic mutations is particularly evident in the establishment of drug tolerance as drug-tolerant states are transient, rapidly emerging, and functionally heterogeneous. The emergence of DTPs has been shown in several studies to involve chromatin modifier enzymes, including bromodomain and extraterminal (BET) proteins, histone lysine demethylases (KDMs), enhancer of zeste homolog 2 (EZH2), and the dynamic remodelling of open chromatin architecture. KDMs have been found to be upregulated in DTPs upon drug treatment of multiple tumors, including lung, breast, and glioblastoma [116,117,118]. Therefore, the use of KDM inhibitors is potentially viewed as a strategy to prevent the emergence of DTPs upon treatment with conventional or targeted drugs. In fact, pretreatment with the specific KDM inhibitor CPI-455 was shown to elevate the global levels of H3K4 trimethylation and decrease DTP numbers in multiple cancer cell lines treated with chemotherapy or targeted agents [117]. More recently, an inhibitor of iron-dependent KDMs was shown to inhibit breast tumor growth in murine xenograft models [119]. Risom et al. found a high degree of cell heterogeneity (assessed as expression of differentiation-state markers) and the emergence of DTPs upon treatment with multiple pathway-targeted compounds in TNBC and basal-like BC. Importantly, epigenetic mechanisms that are responsible for DTPs generation could be counteracted by the PI3K/mTOR inhibitor BEZ235 and the BET inhibitor JQ1, which prevented the acquisition of a DTP state resulting in cell death in vitro and xenograft regression in vivo [120]. BET inhibitors have also been recently shown to revert drug resistance and block pro-tumorigenic activity by disrupting YAP/TAZ binding to the epigenetic coactivator bromodomain-containing protein 4 (BRD4), which controls the expression of multiple growth-promoting genes [121]. In addition to histone methylation changes, DTPs were shown to also rely on alterations of histone deacetylase (HDAC) activity and histone acetylation patterns [108,122,123]. Finally, recent discoveries indicate that transposable elements (TEs), historically overlooked as junk DNA, may be involved in the epigenetic regulation of drug resistance. DTPs have been shown to promote silencing of TEs (and specifically of LINE-1 sequences) in breast, lung, colorectal, and melanoma cancer cell lines through increasing histone H3K9 and H3K27 methylation to survive and protect their genomes in response to drug or stress exposure. According to this hypothesis, disruption of the repressive chromatin over LINE-1 elements resulted in DTPs ablation [123]. Altogether, these results suggest that specific inhibitors of epigenetic regulators can counteract the emergence of DTPs and may find clinical use in preventing disease relapse.

## 7. Role of Microenvironmental Factors in Dictating Breast Cancer Stemness and Chemoresistance

The tumor microenvironment (TME) is a critical driver of the malignant phenotype and it has been shown to be specifically implicated in promoting CSCs functions, such as self-renewal, plasticity, and chemoresistance in breast and other cancers [13]. TME components include cellular elements (such as stromal fibroblasts, adipocytes, immune cells, mesenchymal stromal cells, and endothelial cells), subcellular elements (exosomes and microvesicles), and soluble factors (growth factors, hormones, and cytokines). The extracellular matrix (ECM) and its collagen composition as well as mechanical properties, such as matrix stiffness, also crucially affect cancer stemness. Moreover, chemico-physical parameters of the TME, such as oxygen concentration, pH, and nutrient availability, have been shown to influence breast cancer stemness and tumor aggressiveness (Figure 2). 

In this regard, hypoxic microenvironments were recently shown to generate dormant chemoresistant BC cells and enhance the stem cell phenotype in BC xenografts [102,124,125]. In turn, the CSCs population actively modifies the surrounding TME through cell-cell interactions, cytokine secretion, and extracellular vesicles production, which contributes to creating a tumor-promoting ecosystem [13]. Here, we will focus on recent advancements on BC microenvironment and its role in supporting the BCSCs population since cellular and non-cellular components of the breast TME have been analysed in detail in recent reviews [126,127,128]. Great advancements have been made recently in dissecting the nature and function of TME components in BC with a variety of techniques. Multiplexed imaging was applied to the analysis of TME in TNBC, which provides new insights on tumor-immune cell interaction, spatial arrangement, and regulatory protein expression [129]. Infiltrating immune cells in the BC TME were profiled at the single cell level by RNA-seq and T-cell receptor sequencing, providing new knowledge on T-cell diversity in BC [130]. Analysis of the infiltrating myeloid component in TNBC revealed the existence of immune subtypes that were related to different sensitivity to immune checkpoint blockade [131]. Different from myeloid cells, which are often involved in tumor-promoting effects, tumor-infiltrating B cells have been recently shown to be responsible for the generation of humoral immune responses in BC through the production of cytokines and immunoglobulins, contributing to local anti-tumor immunity [132]. CAFs are the major stromal cells that contribute to the TME. Recently, CAFs in BC were analysed at the single cell level and then categorized in different subclasses with different origin, functional programs, and prognostic value [133]. Another study characterized four CAF subsets in BC with distinct properties and different abilities to influence T-cell survival and differentiation [134]. Notably, CAFs were recently recognized, even as determinants of the BC molecular subtype. In fact, the crosstalk between cancer cells expressing platelet-derived growth factor (PDGF) and CAFs expressing the cognate receptors in human basal-like mammary carcinomas results in a hormone receptor-negative state. Strikingly, interfering with PDGF activity converts cancer cells to a hormone receptor-positive state, which enhances endocrine therapy sensitivity in previously resistant tumors [135]. A specific role of CAFs in supporting BC stemness was reported by Su and coworkers, who distinguished two CAFs subsets on the basis of CD10 and GPR77 expression. The CD10^+^GPR77^+^ CAF subset was linked to chemoresistance and poor prognosis in BC and lung cancer patients and a neutralizing anti-GPR77 antibody abolished tumor formation in PDX models, which indicated CAFs targeting as a therapeutic strategy that is able to affect the BCSCs compartment [136]. An interesting link between BC, TME and inflammation has been recently provided by a study showing that CAFs are able to sense damage-associated molecular patterns and activate the inflammosome pathway, mediated by NLRP3 and IL1β. In turn, CAFs-derived inflammasome promoted tumour progression and metastasis by creating an immune suppressive TME and upregulating the expression of adhesion molecules on endothelial cells [137]. Finally, the influence of TME on BC cell metabolism is receiving increasing attention. Recently, it was shown that extracellular vesicles produced by BC cell contain miR-105, which activates MYC signalling in CAFs, inducing their metabolic reprogramming. MiR-105-reprogrammed CAFs are induced to display different metabolic features in response to changes in the metabolic environment to provide optimal support to tumor growth [138]. Metabolic factors, such as the availability of specific nutrients, were also shown to crucially influence BC interactions with the TME and its ability to metastasize. The availability of pyruvate was essential to drive collagen-based remodelling of the ECM in the lung metastatic niche. In fact, pyruvate induces the production of α-ketoglutarate that activates collagen hydroxylation and stimulates the growth of BC lung metastases in mouse models [139]. Adipocytes are tightly linked with tissue metabolism and they are the object of intense studies, also due to the correlation between BC and obesity. The secretion of GM-CSF and MMP9 by adipose progenitor cells was previously shown to result in local and metastatic BC progression, which could be reverted by GM-CSF neutralization or metformin treatment [140]. Lipid metabolism in BCSCs has been recently demonstrated to be regulated by Janus kinase (JAK)/STAT3, which promotes stemness and chemoresistance. Leptin that is produced by mammary adipocytes also activates STAT3 and downstream fatty acid oxidation, further linking adipose tissue with BCSCs functions [141]. Furthermore, IL8 secreted by cancer-associated adipocytes was shown to play a pro-tumorigenic effect in a STAT3-dependent manner, and the inhibition of the IL8 signaling using specific short hairpin RNA, anti-IL8 antibody, or reparixin may represent an effective therapeutic strategy [142]. Adipsin, a cytokine that is secreted by mammary adipocytes abundant in obese patients, has been indicated as a mediator of stemness in the BC TME, adding evidence to link between BC, stem cells, and obesity [143]. In addition to ubiquitous TME components, organ-specific mature cells have also been shown to influence tumor progression, particularly in the metastatic setting. The interactions between BC and bone marrow cells have been extensively studied [144]. However, bone marrow cells were recently recognized to play an unexpected role in BC aggressiveness. In fact, bone marrow-derived mesenchymal stromal cells (BM-MSCs) were shown to migrate to primary breast tumors and lung metastases and differentiate to a distinct subpopulation of CAFs with tumor-promoting functions and a peculiar PDGFRa-negative phenotype correlated with worse prognosis [145]. BM-MSCs were also shown to be activated in response to interleukin-11 (IL11) produced by a subpopulation of BC cells, which results in the stimulation of pro-tumorigenic and pro-metastatic neutrophils [146]. Other factors that are produced by bone marrow cells, such as osteopontin, further contribute to the crosstalk between BC and the bone niche promoting cancer cell stemness and migratory ability [147]. Besides the bone marrow, novel interactions are also emerging between BCSCs and other components of the pre-metastatic niche, such as brain or lung cells. The lung microenvironment was previously shown to induce the chemotactic migration of BCSCs through CD44 engagement [148]. More recently, a new system that is based on labelling of the local metastatic cellular environment allowed for identifying a population of cancer-associated parenchymal stem cells derived from the lung epithelium and able to support BC cells [149]. In the premetastatic brain niche, a truncated form of GLI1 was recently reported to promote BCSCs self-renewal by activating the transcription of stemness genes CD44, NANOG, SOX2, and OCT4. Furthermore, BCSCs expressing truncated GLI1 strongly activated and interacted with astrocytes, leading to brain metastasis [150]. Interactions between BCSCs and the TME have huge clinical implications and they could be the key for effective therapeutic strategies. As previously mentioned, the SMO inhibitor Sonidegib provided a moderate benefit in a phase I clinical trial on patients with metastatic TNBC, providing a proof-of-principle of the efficacy of targeting CAFs signalling [52]. In summary, emerging evidences indicate that targeting TME factors could have a beneficial effect on tumor progression by disrupting interactions that are essential for BCSCs survival. However, evaluations of TME-targeting agents should keep in mind that the mouse and the human breast microenvironments are profoundly different [151]. Therefore it is important to use suitable models, such as humanized orthotopic xenografts, patient-derived organoid cultures, or artificial niches [152,153,154], which more accurately recapitulate the human breast microenvironment.

## 8. Disseminated Tumor Cells: Dormant BCSCs in Premetastatic Niches

In contrast to the traditional notion that metastasis is a late event in tumor progression, increasing evidences indicate that neoplastic cells can disseminate from early tumors, which results in the parallel progression of primary tumor and metastases [155]. The concept of early metastatic dissemination is supported by gene expression studies performed in breast and prostate cancer, showing that distinct genetic alterations occur in primary tumors and metastatic cells [156,157,158]. In BC patients without evident metastasis, approximately 50% of cytokeratin-positive cells that were isolated from the bone marrow show less chromosomal alterations than primary tumors, which indicates that they disseminated before the occurrence of genomic instability events [156]. Similar evidences were collected by comparing DTCs that were isolated from patients after curative resection of the primary tumor (M0) with overt metastatic cells (M1) and primary tumor cells. M0 cells displayed significantly fewer chromosomal aberrations than primary tumors or M1 cells and their aberrations appeared to be randomly generated, which indicated that tumor cells disseminate in a precocious genomic state [157]. In HER2/PyMT transgenic mouse and ductal carcinoma in situ (DCIS) patients, DTCs were found in the bone marrow and lungs before the primary tumor became morphologically invasive [46,159,160]. Notably, early metastatic cells have been shown to possess the combined features of stemness and dormancy. Lawson et al. compared BC cells that were isolated from tissues with low versus high metastatic burden and found that cells from low-burden tissues displayed an increased expression of stemness, EMT, pro-survival, and dormancy-associated genes, appearing as quiescent/slow cycling BCSCs. By contrast, metastatic cells from high-burden tissues were similar to primary tumour cells, which were more heterogeneous and expressed higher levels of differentiation genes [161]. In a HER2-driven mouse BC model, Hosseini and coworkers demonstrated that the cells from early lesions displayed enhanced stemness, migratory, and prometastatic features as compared to cells from advanced tumors and that an impressive 80% of metastases were derived from early disseminated cancer cells [160]. Similarly, in early breast tumors HER2 was found to activate a WNT-dependent EMT-like dissemination program without the complete loss of the epithelial phenotype, which was reversed by HER2 or WNT inhibition [46]. Altogether, these studies depict a scenario in which metastatic dissemination is an early step during tumorigenesis, being necessary but not sufficient for metastatic outgrowth. The interaction of DTCs with the ectopic microenvironment then leads to a selection and/or adaptation within the early metastatic niche, which eventually leads to the acquisition of a distinct genetic profile [159]. During this process, quiescence signalling pathways that are related to TGFβ and p38 have been shown to be essential for DTCs survival and resistance to the foreign environment [162]. Emerging evidences also suggest the existence of organ-specific niches that promote DTCs survival, protecting them from microenvironmental stress and therapy-related toxicity [163,164]. As a part of the premetastatic microenvironment, immune system cells have been shown to control DTCs dormancy [165] and immunotherapy has been proposed as a potential strategy for eradicating DTCs [166]. Strong alterations to homeostasis represented by perioperative surgery or inflammation have been demonstrated to disrupt DTC dormancy and support metastatic outgrowth in several tumors [158,167,168,169]. On the other side, anti-inflammatory agents, such as nonsteroidal anti-inflammatory drugs (NSAIDs), seem to dramatically decrease the risk of metastatic relapse, possibly by preventing the reawakening of dormant cells caused by niche alterations that occur during inflammation [170,171]. The ability of dormant DTCs to persist in a quiescent state or to resume proliferation might result from both microenvironmental signals and epigenetic reprogramming mechanism. Early studies discovered that the balance between proliferation and dormancy is determined by the ratio between the activity of p38 and ERK1/2 [172]. Further insights into p38-activated pathways led to the key discovery that the orphan nuclear receptor NR2F1 upon activation by p38 induces dormancy through SOX9, RARβ, CDK inhibitors, and global chromatin repression [173]. Recently, NR2F1 has emerged as a clinical marker of dormancy in BC, with its expression being able to discriminate patients with short term systemic relapse from those with long disease-free intervals [174]. Mitogen- and stress-activated kinase 1 (MSK1) was recently identified as an important marker and regulator of metastatic dormancy in ER^+^ BC patients, indicating that stratifying patients according to MSK1 expression could improve prognosis [175]. Collectively, studies on DTPs biology support a view where the dormancy window could be exploited for therapeutic purposes. Clinical trials that aimed at targeting premetastatic DTCs [166] will likely impact the future management of BC patients.

## 9. Clinical Relevance of Quiescent BCSCs and Potential Therapeutic Strategies

Understanding the biology of dormant CSCs has a great clinical relevance for the treatment of BC patients. In fact, while more aggressive forms of BC, such as TNBC, have a peak of distant recurrence within three years of diagnosis [176], ER^+^ BC patients have a risk of late relapse until twenty years from primary tumor removal [177]. Such a long period of latency indicates that DTCs that are present in premetastatic organs implement a series of survival strategies that allow them to remain alive and dormant while also maintaining their tumorigenic features and escaping immune surveillance. Kim et al. generated a 49-gene signature for tumor cell dormancy and reported that disseminated ER^+^ tumor cells carrying such a signature were more likely to undergo prolonged dormancy before resuming metastatic growth. Moreover, the suppression of dormancy-associated genes basic helix-loop-helix family member 41 (BHLHE41) and NR2F1 resulted in increased in vivo growth of ER^+^ MCF7 cells, highlighting the importance of dormancy regulation for BC progression and relapse [178]. Therapeutic approaches dealing with dormant BCSCs and aiming at the prevention of tumor relapse consist in so-called sleeping, killing, and awakening strategies (Figure 3) [179]. Sleeping strategies are aimed at maintaining DTCs in a harmless dormant state and they are based on drugs that suppress signals required for BCSCs proliferation and/or survival. Among them, anti-estrogen therapies and CDK4/6 inhibitors are used, respectively, to prevent relapse and to inhibit metastasis progression in ER^+^ BC [180,181].

Adjuvant anti-estrogen therapies given for up to ten years after diagnosis have significantly improved the survival of patient with ER^+^ BC [181]. The maintenance of dormancy has also been experimentally achieved by increasing the expression of dormancy-related factors, including p38 MAPK [172], DYRK1A [182,183], and N2RF1 [174,178]. Sleeping strategies have several drawbacks despite their great potential for the clinical management of BC patients. First, some premetastatic tumor cells may not respond to drug treatment or they may develop resistance with time, resulting in disease recurrence, as happens for approximately 20% of BC patients treated with anti-estrogen therapies [34]. Secondly, dormancy-inducing drugs must be taken for long periods of time (ideally for lifetime) and are therefore associated with intolerance to side effects, problems of patient compliance, and high costs. In this regard, novel fenretinide derivatives have been reported to possess a broad range antitumor activity and the ability to neutralize the CSCs compartment through the combined activation of antimetabolic, apoptotic, and dormancy-promoting pathways [184,185]. These properties, which are associated to an enhanced bioavailability, affordable cost, and low toxicity may candidate new fenretinide formulations as potential dormancy-inducing drugs in BC. Awakening strategies are designed to reactivate the cell cycle in dormant tumor cells, which would theoretically become more vulnerable to cytotoxic chemotherapy. In BC, the inhibition of FBXW7 has been shown to induce DTCs exit from the quiescent state, thereby sensitizing them to conventional chemotherapy [186]. However, awakening strategies may be more feasible in hematologic cancers than in solid tumors. In fact, in the latter, they represent a risky approach, as DTCs that are awakened from dormancy may acquire the features of highly aggressive and therapy resistant metastatic cells. The third strategy consist in eliminating DTCs while dormant. In BC, the simultaneous inhibition of SRC signalling and MEK1/2-ERK1/2 signalling has been reported to induce the apoptosis of dormant cells [187]. Selective KDM5 inhibitors have been shown to eradicate dormant DTPs in lung, melanoma, and BC models [117]. Recently, the activation of ferroptosis via the inhibition of GPX4 was proposed as a strategy to kill DTPs in BC and other cancers [115,188]. However, killing strategies may also fail to completely eradicate dormant cancer cells due to BCSCs heterogeneity and plasticity that may drive the emergence of resistant tumor cells. A recent provocative study challenged the role of dormancy-related signals in the regulation of premetastatic states, showing that BC DTCs are protected from chemotherapy through integrin-mediated interactions, irrespective from their cell cycle status. Disrupting the interactions between DTCs and the perivascular niche with integrin inhibitors resulted in DTCs chemosensitization and they may represent a novel therapeutic approach to eradicate these cells [189]. Besides pharmacologic strategies directed against BCSCs, new therapeutic approaches that are based on the combination of immunotherapy with targeted agents are being explored as an opportunity for BC patients, even in the metastatic setting [190]. Moreover, it is likely that a variety of micro- and macroenvironmental signals influence DTCs biology and, consequently, their alternative sleeping/awakening fate. Indeed, lung inflammation (that can be induced by tobacco smoke) has been shown to awaken premetastatic cells in breast and other tumor models [167,168,169]. Neutrophils are primarily implicated in the regulation of premetastatic cells that are located in the lungs. In fact, in steady-state conditions neutrophils produce intact functional thrombospondin-1 that confers a tumor-inhibitory microenvironment. However, under inflammatory conditions the same neutrophils degranulate and release elastases and proteases, which creates a metastasis-supportive microenvironment [169]. The mechanism of neutrophil-mediated awakening of dormant cells has been further elucidated by demonstrating that, in the presence of bacteria or tobacco-induced inflammation, neutrophils generate scaffolds of DNA with associated elastases, proteases, and matrix metalloproteinase, called Neutrophil Extracellular Traps (NETs). NETs induce the proteolytic remodeling of laminin, thus revealing an epitope that triggers the proliferation of dormant cancer cells through integrin activation and FAK/ERK/MLCK/YAP signaling [167]. These observations also confirm the emerging importance of integrins in promoting cancer cell stemness and therapy resistance [191]. Besides inflammation and infections, a number of microenvironmental changes that are produced by lifestyle-related factors could influence dormant DTCs by increasing or decreasing their probability to awaken [192]. Obesity and high fat diet have been demonstrated to increase the risk of BC recurrence [193,194,195], although the mechanisms that are responsible for this phenomenon are largely unknown. On the other side, low calories fasting-mimicking diets and physical exercise are emerging as important factors influencing the tumor microenvironment and, therefore, they could be exploited not in only cancer prevention, but also to influence therapy response and metastatic recurrence [196,197].

## 10. Conclusions

Although great progress has been made over the past twenty years in BC diagnosis and treatment, BC remains a potentially deadly disease due to the ability of BCSCs to resist therapies, disseminate, and metastasize. The ongoing advancements of basic and preclinical research in understanding BCSCs biology must be readily translated into clinical practice to improve the effectiveness of current cancer therapies. Moreover, the emerging network existing between BCSCs, the immune system, and the micro/macroenvironment will likely provide essential information regarding how to integrate anticancer therapies and lifestyle-related factors for improved management of BC patients.

## Figures and Tables

**Figure 1 cancers-11-01569-f001:**
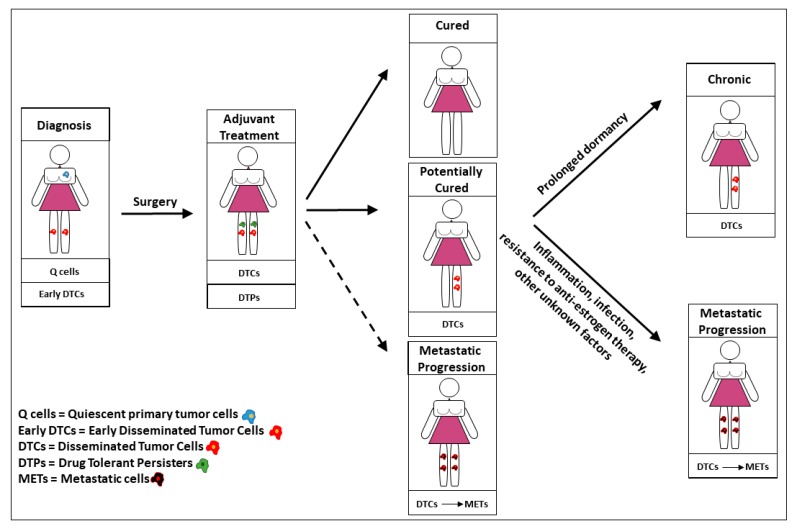
Role of quiescent/dormant breast cancer stem cells in patient outcome. Quiescent breast cancer stem cells are present in primary tumors particularly in hypoxic areas and can disseminate early during tumor progression. Chemotherapy may induce the formation of drug-tolerant persisters (DTPs) which survive treatment as well as dormant disseminated tumor cells (DTCs). Patient outcome is crucially determined by the fate of DTCs: patients are totally cured if dormant cells are eradicated by therapies, or potentially cured if DTCs remain alive and dormant. In potentially cured patients, DTCs can remain dormant throughout lifetime (resulting in a chronic but harmless presence of cancer) or awaken following inflammations, infections, emergence of endocrine therapy resistance, and generate metastatic tumours.

**Figure 2 cancers-11-01569-f002:**
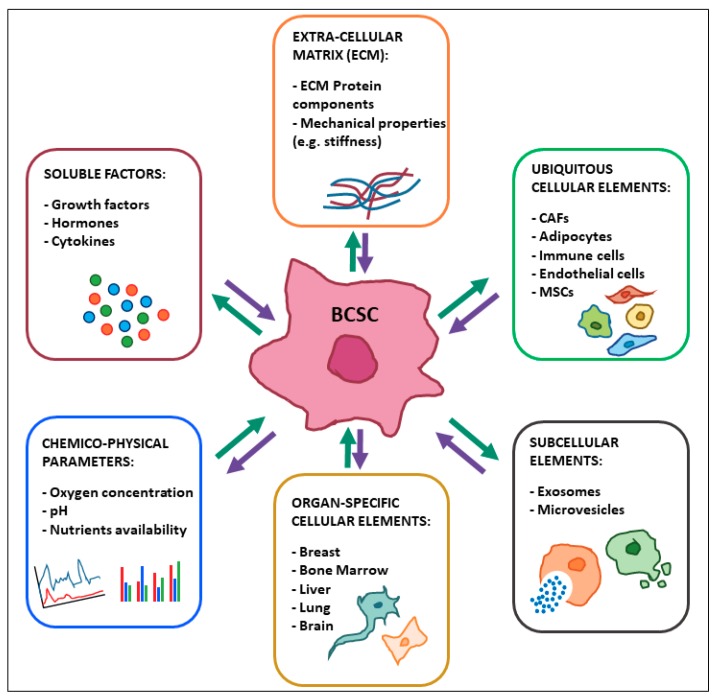
Crosstalk between breast cancer stem cells (BCSCs) and microenvironmental factors. CAFs: Cancer-Associated Fibroblasts; MSCs: Mesenchymal Stromal Cells.

**Figure 3 cancers-11-01569-f003:**
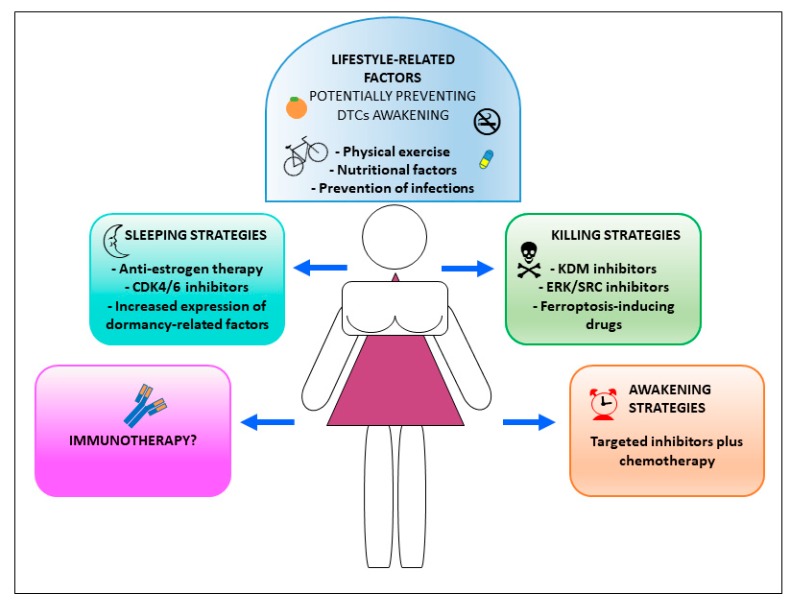
Therapeutic strategies and factors related to DTCs dormancy in breast cancer (BC) patients. Distinct therapeutic strategies and lifestyle habits cooperate to prevent tumor relapse in BC patients.

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
