# Peer review of "Breast Cancer Stem Cells as Drivers of Tumor Chemoresistance, Dormancy and Relapse: New Challenges and Therapeutic Opportunities"

_cancers, 2019, doi:10.3390/cancers11101569_

Round 1

Reviewer 1 Report

The manuscript is well written and covers important aspects of breast cancer stemness plasticity and therapeutic resistance. The most relevant articles have been included as reference.

Due to previous publications on the same topic (PMID: 29506506; PMID: 31430951; PMID: 30834247; PMID: 31355143) the originality of the article may be a concern. However, the article can still stand out because of it breast cancer specificity. Some aspects of the current version need improvement.

Include a section describing the known differences/similarities between breast cancer stem cells and other cancer stem cells (Lung, brain, skin, kidney, colon…) Improve graphical representations. Understanding the correlation between BCSCs, dormancy and clinical outcome of patients is less challenging. The authors would rather summaries the majors mechanisms involved in stem cell formation, maintenance, and differentiation in primary as well as metastatic breast cancer. Similarly, the role of the tumor microenvironment (CAFs, immune cells…) in breast cancer stemness must be summarized by a graphical representation. Therapeutic options can also be integrated.

Author Response

We would like to thank the Reviewer for his/her positive evaluation of the manuscript and for helpful suggestions. We have made the following modifications in response to the Reviewer’s requests:

We have added an extensive section concerning the role of the tumor microenvironment (TME) in modulating breast cancer and stem cells. This new section is based on very recent articles focused on breast cancer microenvironment and, in our opinion, adds significant value and originality to the work. We have added a new figure (Figure 2 in the revised version) that summarizes the role of the TME in breast cancer stemness. New TME-related therapeutic options have been integrated in the revised manuscript. We have improved the graphical representation of all the figures. We did not include an additional section concerning the differences between breast cancer CSCs and CSCs of other tumors, nor concerning the mechanisms involved in stem cell formation, maintenance and differentiation in primary and metastatic tumors. In fact, the first issue in our opinion was not instrumental to the topic of this review, while the second request was exceedingly burdensome and time-consuming, considering that the review is already very comprehensive, with approximately 200 references, and the time allowed for minor revisions is very limited.

Reviewer 2 Report

Quiescent and dormant CSCs root the treatment resistance and later on relapse of tumors. This review article provides a summary on the current understanding of breast CSC dormancy, which is useful information for researchers.

Author Response

We thank the Reviewer for his/her appreciation of the manuscript.

Reviewer 3 Report

De Angelis et al. present a review covering breast cancer stem cells (BCSCs). The review places emphasis on dormant cells in chemoresistance, dissemination, and metastasis. Overall, it is a comprehensive compilation of studies and touches on interesting aspects of disseminated tumor cells that are dormant along with therapeutic strategies to overcome these cells. The review is well structured and informative, some minor suggestions are listed as follows:

In section 2, it would be beneficial to illustrate the potential of single cell omics platforms, to better define BCSC states and their heterogeneity. (PMID: 29681456, 30181541, 29120751). Line 35, change “in” to “into” Line 86, change “reflexes” to “reflects” Line 456, change “from ALDH” to “on ALDH”

Author Response

We would like to thank the Reviewer for his/her positive evaluation of the manuscript and for helpful suggestions. We have made the following modifications in response to the Reviewer’s requests:

We added a paragraph in the Introduction to illustrate the potential of single cells omics platform in dealing with breast cancer heterogeneity and we cited the excellent papers indicated by the Reviewer. We have made the spelling corrections as indicated.